# Neutronic Characteristics of ENDF/B-VIII.0 Compared to ENDF/B-VII.1 for Light-Water Reactor Analysis

**Kang-Seog Kim \* and William A. Wieselquist**

Nuclear Energy and Fuel Cycle Division, Oak Ridge National Laboratory, One Bethel Valley Road, P.O. Box 2008, MS-6172, Oak Ridge, TN 37831, USA; wieselquiswa@ornl.gov

**\*** Correspondence: kimk1@ornl.gov

**Abstract:** The Evaluated Nuclear Data File (ENDF)/B-VIII.0 data library was released in 2018. To assess the new data during development and shortly after release, many validation calculations were performed with static, room-temperature benchmarks. Recently, when performing validation of ENDF/B-VIII.0 for pressurized water reactor depletion calculations, a regression in performance compared to ENDF/B-VII.1 was observed. This paper documents extensive benchmark calculations for light-water reactors performed using continuous energy Monte Carlo code with ENDF/B-VII.1 and -VIII.0 and neutronic characteristics of ENDF/B-VIII.0 are discussed and compared to those of ENDF/B-VII.1. It is our recommendation that ENDF/B data library assessment should include reactor-specific benchmark assessments, including depletion cases, such that these types of regressions may be caught earlier in the data development cycle.

**Keywords:** ENDF/B-VII.1 and VIII.0; light water reactor; neutronic characteristics; benchmark calculation

## 1. Introduction

The Virtual Environment for Reactor Applications (VERA) [1], developed by the Consortium for Advanced Simulation of Light Water Reactors (CASL) [2], was very successful in simulating high-fidelity multiphysics for pressurized water reactor (PWR) physics analysis. VERA development continues through the Nuclear Energy Advanced Modeling and Simulation (NEAMS) project [3] for the boiling water reactor (BWR) analysis. The PWR simulation results using VERA with the Evaluated Nuclear Data File (ENDF)/B-VII.1 nuclear evaluated data [4] were very consistent with the PWR measured data [5], which were processed to generate the 51-group neutron cross-section library [6,7] for the primary VERA neutronics simulator MPACT [8]. The ENDF/B-VII.1 MPACT 51-group library supports both non-epithermal and epithermal upscattering for $^{238}U$, which has been known to improve the Doppler temperature coefficient [9]. However, the benchmark calculations in the paper by Mangham et al. [5] were performed without considering epithermal upscattering, because epithermal upscattering would result in about a 200 pcm reactivity underestimation. It was expected that the ENDF/B-VIII.0 data library [10] would resolve the reactivity bias issue by considering epithermal upscattering for the PWR analysis. Extensive benchmark calculations were performed using ENDF/B-VIII.0 to identify neutronic characteristics of ENDF/B-VIII.0 and results are compared to ENDF/B-VII.1.

ENDF/B-VIII.0 was released in 2018. Identifying potential issues in new ENDF/B libraries for power plant simulations can be challenging because most validation calculations for new versions are performed for static and room-temperature benchmark problems. Depletion benchmark calculations have been performed to investigate the differences between ENDF/B-VII.1 and -VIII.0 [11] and results have shown that ENDF/B-VIII.0 significantly underestimated reactivity compared to ENDF/B-VII.1 with respect to depletion. Park et al. [12] compared both libraries in the VERA benchmark progression problems and a similar trend to that discussed in the paper by Kim and Wieselquist [11] was indicated.

However, the most influential nuclides were not identified and insufficient information was included for the neutronic behavior of ENDF/B-VIII.0 in terms of light-water reactor (LWR) physics analysis. Bostelmann et al. also demonstrated the impact of ENDF/B-VIII.0 on advanced reactor simulation [13], but the investigation concerned only its basic impact on various advanced reactor analysis.

CASL and NEAMS developed the PWR and BWR benchmark problems to verify the MPACT multigroup (MG) library for VERA, covering most neutronic characteristics for the PWR and BWR physics analysis. The reference solutions were obtained using the continuous-energy (CE) Monte Carlo (MC) codes SCALE/KENO [14], Serpent [15] and Monte Carlo N-Particle (MCNP) Transport [16] with ENDF/B-VII.1. In addition to the depletion comparison between ENDF/B-VII.1 and -VIII.0 [11], benchmark calculations were performed using the CE MC codes to identify the neutronic characteristics of ENDF/B-VIII.0 compared to those of ENDF/B-VII.1 for LWR physics analysis.

This study focuses on comparisons between ENDF/B-VIII.0 and -VII.1 for extensive PWR and BWR benchmark problems to clearly identify the impact of ENDF/B-VIII.0 on LWR analysis. A full investigation will be completed during the development of new versions of ENDF/B through a collaboration with nuclear data evaluators within the Cross Section Evaluation Working Group and it is expected to significantly shorten the ENDF/B development period and minimize potential issues.

## 2. LWR Benchmark Problems

### 2.1. PWR Benchmark Problems

The PWR benchmark problems were taken from Kim et al. [7], which verified the ENDF/B-VII.1 MPACT 51-group library for the VERA MPACT. Only a brief overview of the problems is included herein; detailed descriptions can be found in the aforementioned paper [7]. These benchmark problems were used in the comparison of neutronic characteristics between ENDF/B-VII.1 and -VIII.0. Table 1 provides a summary of the PWR benchmark suite.

**Table 1.** The PWR benchmark suite.

| Case | Description | Number of Cases | Reference |
|------|-------------|-----------------|-----------|
| A | VERA core physics progression problems | 26 | [7,17] |
| B | Extended VERA progression problems | 15 | [7] |
| C | VERA depletion problems | 24 | [7,18] |
| D | Extensive PWR pin and assembly problems | 1620 | [7,19,20] |
| E | Nonuniform fuel temperature problems | 14 | [21] |
| F | Mosteller benchmark problems | 21 | [22] |

The VERA core physics benchmark progression problems were created to provide a method for developing and demonstrating new reactor physics methods and software capabilities [17]. The progression problems range from a simple 2D pin cell to the full-cycle depletion and refueling of a 3D reactor core configuration with control rods and burnable poisons consistent with actual nuclear power plant designs. Most of the data are based on fuel and plant data from the initial core loading of Watts Bar Nuclear Unit 1, a Westinghouse Electric Company-designed 17 × 17 PWR.

Because the VERA core physics benchmark progression problems do not include various $^{235}$U enrichment and burnup compositions, additional benchmark problems were developed to analyze the cross-section library's sensitivities to $^{235}$U enrichment, burnup and a number of radial fuel rings. These benchmark problems were developed based on the VERA progression problems 1B and 1C.

The VERA depletion benchmark problems [7,18] were developed based on the VERA progression problems. The depletion benchmark suite includes 8 single-pin problems and 16 fuel-assembly problems with various fuel temperatures, $^{235}$U enrichments, control rods and burnable poisons.

The extensive benchmark problems for PWR fuel pins and assemblies were developed to determine how well the MPACT MG library agrees with CE MC results [19,20]. The PWR fuel pin cases consist of three [235]U enrichments, four rod sizes, three hot coolant densities, three hot fuel temperatures, cold cases at room temperature and three boron concentrations. There is a total of 360 cases, 324 hot cases and 36 cold cases. The PWR pin cells were modeled as three concentric rings of fuel, helium gap and zirconium, surrounded by a square region of coolant. Additional benchmark problems were developed for 14 PWR assembly types, including $15 \times 15$, $16 \times 16$ and $17 \times 17$ designs by different fuel vendors and these benchmarks account for many state-point conditions encountered in a reactor. Each assembly type includes 90 state points with three [235]U enrichments, three hot coolant densities, three hot fuel temperatures, one cold case at standard room temperature and density and three boron concentrations. There is a total of 81 hot cases and 9 cold cases per assembly, for a total of 1260 cases.

Seoul National University developed a benchmark suite for the intrapellet nonuniform temperature distribution cases [21]. The geometrical specifications include five equivolume subzones in the fuel pellet, gap, cladding and moderator. Nonuniform temperature profiles as a function of power and average fuel temperatures were introduced. Benchmark calculations were performed using both nonuniform and uniform temperature profiles.

The Mosteller benchmark [22] was originally developed to measure the Doppler temperature reactivity coefficient and has also been used for validation of software. The original benchmarks did not include the 1200 K cases; those cases were added to these benchmark calculations.

### 2.2. BWR Benchmark Problems

### 2.2.1. VERA BWR Depletion Benchmark Problems

The VERA BWR depletion problems were developed for single-pin fuels only. Single-pin benchmarks (shown in Table 2) were developed with fuel pins of a Peach Bottom 2 (PB2) $7 \times 7$ design with a large fuel pellet radius. Additionally, 50% and 70% voids were considered to observe the trend of reactivity difference for various void fractions.

**Table 2.** Benchmark cases for the BWR VERA depletion benchmarks.

| No. | Fuel | Type | Pellet Radius (cm) | [235]U wt % | Temperature (K) | | | Void (%) | Power Density (w/gU) |
|-----|------|------|-----|-----|------|------|------|-----|-----|
| | | | | | Fuel | Clad | Mod. | | |
| 1A | Pin | PB2_7×7 | 0.60579 | 3.1 | 600 | 600 | 600 | 0 | 40 |
| 1B | Pin | PB2_7×7 | 0.60579 | 3.1 | 900 | 600 | 600 | 0 | 40 |
| 1C | Pin | PB2_7×7 | 0.60579 | 3.1 | 1200 | 600 | 600 | 0 | 40 |
| 1D | Pin | PB2_7×7 | 0.60579 | 2.1 | 900 | 600 | 600 | 0 | 40 |
| 1E | Pin | PB2_7×7 | 0.60579 | 3.6 | 900 | 600 | 600 | 0 | 40 |
| 1F | Pin | PB2_7×7 | 0.60579 | 4.6 | 900 | 600 | 600 | 0 | 40 |
| 1G | Pin | PB2_7×7 | 0.60579 | 3.1 | 900 | 600 | 600 | 50 | 40 |
| 1H | Pin | PB2_7×7 | 0.60579 | 3.1 | 900 | 600 | 600 | 70 | 40 |

### 2.2.2. Extensive BWR Pin and Assembly Benchmark Problems

Single-pin benchmark problems were developed using four BWR fuel types [23], as shown in Table 3. The benchmark problems include four fuel pellet sizes, four moderator/fuel temperatures, three [235]U enrichments, four void fractions and four burnups. The analysis used a total of 220 benchmark cases.

**Table 3.** Benchmark cases for the BWR single pins.

| Category | Case | Specification | Cases | Total | Remarks |
|---|---|---|---|---|---|
| Fuel types, fuel radius/pitch (cm) | PB Type-6 GE9 GE14 PB2 7 × 7 | 0.52070/1.62560 0.53213/1.62560 0.43800/1.30000 0.60579/1.87452 | 4 | | - |
| Moderator/fuel temperatures (K) | CZP [a] HZP [b] HFP [c]-1 HFP-2 | 293/293 600/600 600/900 600/1200 | 4 | 156 | Zero burnup |
| $^{235}$U enrichment (wt %) | - | 2.1/3.1/4.1 | 3 | | - |
| Void fraction (%) | - | 0/50/70/90 | 4 | | 0% void only for CZP |
| Burnup (MWD/kgU) | - | 10/20/40/60 | 4 | 64 | 3.1 wt%, HFP-1, all voids and all fuel types |
| All | - | - | - | 220 | - |

[a] CZP, cold zero power; [b] HZP, hot zero power; [c] HFP, hot full power.

Extensive BWR assembly benchmark problems were developed based on the VERA BWR progression problems [23], with four fuel pellet sizes, four $^{235}$U enrichments, four moderator and fuel temperatures, three void fractions and control rod in and out. Table 4 provides variations of nuclear state parameters for these benchmark problems. The total number of cases is 320.

**Table 4.** Benchmark cases for the BWR single assemblies.

| Category | Case | Specification | Cases | Total | Remarks |
|---|---|---|---|---|---|
| Fuel types, fuel radius/pitch (cm) | PB-T6 GE-09 GE-14 GE-14v | 0.52070/1.62560 0.53213/1.62560 0.43800/1.30000 0.60579/1.87452 | 4 | | - |
| $^{235}$U enrichment (wt %) | - | Mixed/2.1/3.1/4.1 | 4 | | - |
| Control rods | - | Out/In | 2 | 320 | - |
| Moderator/fuel temperatures (K) | CZP [a] HZP [b] HFP [c]-1 HFP-2 | 293/293 600/600 600/900 600/1200 | 4 | | - |
| Void fraction (%) | - | 0/40/80 | 3 | | 0% void only for CZP |
| All | - | - | - | 320 | - |

[a] CZP, cold zero power; [b] HZP, hot zero power; [c] HFP, hot full power.

## 3. Results

### 3.1. PWR Benchmark Results

3.1.1. VERA PWR Benchmark Progression Problems

Table 5 compares the multiplication factors between ENDF/B-VII.1 and -VIII.0 with and without Doppler broadening rejection correction (DBRC) [24], which is used to consider epithermal upscattering. The benchmark calculations were performed using SCALE/KENO with ENDF/B-VII.1 and -VIII.0. The ENDF/B-VIII.0 library overestimates

reactivity for the following cases that include $^{10}$B and Gd isotopes: (a) 1E, 2L, 2M and 2N with the integral fuel burnable absorber (IFBA) burnable poisons; (b) 2E, 2F and 2H with the Pyrex burnable poisons; (c) 2G with the AgInCd control rod insertion; (d) 2O and 2P with the gadolinia rods. The reactivity differences between ENDF/B-VII.1 and -VIII.0 with DBRC and without DBRC are similar. Cross sections of $^{10}$B and Gd isotopes that cause positive reactivity in ENDF/B-VIII.0 are discussed in Section 4.

**Table 5.** Results of the VERA PWR benchmark progression problems.

| Case | KENO with DBRC | | | KENO without DBRC | | |
| --- | --- | --- | --- | --- | --- | --- |
| | $k_{\mathrm{eff}}$ * | | (2–1) $\Delta\rho$ (pcm) | $k_{\mathrm{eff}}$ * | | (2–1) $\Delta\rho$ (pcm) |
| | **VII.1 (1)** | **VIII.0 (2)** | | **VII.1 (1)** | **VIII.0 (2)** | |
| 1A | 1.18569 | 1.18521 | −34 | 1.18700 | 1.18667 | −24 |
| 1B | 1.18065 | 1.18002 | −45 | 1.18214 | 1.18126 | −63 |
| 1C | 1.16895 | 1.16853 | −31 | 1.17144 | 1.17116 | −21 |
| 1D | 1.15885 | 1.15866 | −14 | 1.16258 | 1.16249 | −7 |
| 1E | 0.77082 | 0.77359 | 465 | 0.77127 | 0.77437 | 519 |
| 2A | 1.18081 | 1.18076 | −3 | 1.18187 | 1.18167 | −14 |
| 2B | 1.18190 | 1.18177 | −9 | 1.18323 | 1.18302 | −15 |
| 2C | 1.17125 | 1.17143 | 13 | 1.17362 | 1.17354 | −6 |
| 2D | 1.16189 | 1.16222 | 24 | 1.16556 | 1.16567 | 8 |
| 2E | 1.06829 | 1.06901 | 63 | 1.06953 | 1.07001 | 42 |
| 2F | 0.97462 | 0.97579 | 123 | 0.97569 | 0.97670 | 107 |
| 2G | 0.84674 | 0.84809 | 188 | 0.84766 | 0.84896 | 181 |
| 2H | 0.78705 | 0.78800 | 153 | 0.78793 | 0.78852 | 95 |
| 2I | 1.17865 | 1.17864 | 0 | 1.17962 | 1.17951 | −8 |
| 2J | 0.97378 | 0.97513 | 142 | 0.97496 | 0.97630 | 141 |
| 2K | 1.01864 | 1.01930 | 64 | 1.01977 | 1.02029 | 50 |
| 2L | 1.01760 | 1.01912 | 147 | 1.01868 | 1.02017 | 143 |
| 2M | 0.93778 | 0.94003 | 255 | 0.93855 | 0.94090 | 266 |
| 2N | 0.86840 | 0.87043 | 268 | 0.86944 | 0.87133 | 250 |
| 2O | 1.04613 | 1.04738 | 114 | 1.04717 | 1.04822 | 96 |
| 2P | 0.92566 | 0.92771 | 239 | 0.92670 | 0.92862 | 223 |

* Maximum standard deviation = 0.00013.

### 3.1.2. VERA PWR Extended Benchmark Progression Problems

The VERA PWR extended benchmark suite was used to discern the trend of the multiplication factors with various $^{235}$U enrichments and various snapshot burnups in which atomic number densities of depleted isotopic inventories were taken from single fuel-pin depletion calculation. The benchmark calculations were performed using SCALE/KENO with ENDF/B-VII.1 and VIII.0 of which the multiplication factors are compared in Table 6. ENDF/B-VIII.0 overestimated reactivity at 2.1 wt % of $^{235}$U. However, as $^{235}$U enrichment increased, ENDF/B-VIII.0 underestimated reactivity, resulting in a $\Delta\rho$ bias of about 300 pcm between 2.1 and 4.6 wt % of $^{235}$U. When all heavy and fission product yield nuclides were included, reactivity differences between ENDF/B-VII.1 and VIII.0 were negligible. There were significant error cancelations among $^{235}$U, $^{238}$U, $^{16}$O and $^{239}$Pu, which are discussed in Section 4. The burnup history effect is discussed in Section 3.1.3.

**Table 6.** Results of VERA PWR extended benchmark progression problems.

| Case | KENO with DBRC | | | KENO without DBRC | | |
|---|---|---|---|---|---|---|
| | $k_{eff}$ | | **(2–1)** $\Delta\rho$ **(pcm)** | $k_{eff}$ | | **(2–1)** $\Delta\rho$ **(pcm)** |
| | **VII.1 (1)** | **VIII.0 (2)** | | **VII.1 (1)** | **VIII.0 (2)** | |
| 1B-21 | 1.06871 | 1.06994 | 108 | 1.07002 | 1.07163 | 140 |
| 1B-26 | 1.13385 | 1.13415 | 23 | 1.13548 | 1.13560 | 9 |
| 1B-31 | 1.18048 | 1.18019 | −21 | 1.18211 | 1.18170 | −29 |
| 1B-36 | 1.21951 | 1.21855 | −65 | 1.22096 | 1.21980 | −78 |
| 1B-41 | 1.25125 | 1.24909 | −138 | 1.25244 | 1.25044 | −128 |
| 1B-46 | 1.27712 | 1.27472 | −147 | 1.27871 | 1.27595 | −169 |
| 1C-00-3a | 1.24435 | 1.24341 | −61 | 1.24720 | 1.24587 | −86 |
| 1C-10-3a | 1.08484 | 1.08479 | −4 | 1.08738 | 1.08699 | −33 |
| 1C-20-3a | 1.00059 | 1.00113 | 54 | 1.00297 | 1.00292 | −5 |
| 1C-40-3a | 0.88112 | 0.88135 | 30 | 0.88297 | 0.88318 | 27 |
| 1C-60-3a | 0.80711 | 0.80721 | 15 | 0.80869 | 0.80886 | 26 |
| 1C-10-1h | 1.17128 | 1.17082 | −34 | 1.17394 | 1.17320 | −54 |
| 1C-20-1h | 1.11417 | 1.11401 | −13 | 1.11647 | 1.11657 | 8 |
| 1C-40-1h | 1.03382 | 1.03463 | 76 | 1.03614 | 1.03682 | 63 |
| 1C-60-1h | 0.98625 | 0.98725 | 103 | 0.98849 | 0.98941 | 94 |

### 3.1.3. VERA PWR Depletion Benchmark Problems

Kim and Wieselquist [11] discussed the reactivity underestimation of ENDF/B-VIII.0 compared to that of ENDF/B-VII.1 for the VERA PWR depletion benchmark problems. Depletion benchmark calculations were performed using Serpent with ENDF/B-VII.1 and -VIII.0 ACE format libraries for the VERA depletion benchmark problems. The ENDF/B-VII.1 ACE format libraries were from MCNP6.1 [16] and the ENDF/B-VIII.0 ACE format libraries were processed using NJOY-2016 [25]. Figure 1a compares the multiplication factors ($k_{eff}$) between the two ENDF/B versions without considering epithermal upscattering as a function of burnup, of which standard deviations were about 20 pcm. ENDF/B-VIII.0 significantly underestimated the multiplication factors at high burnup points compared to ENDF/B-VII.1. Figure 1b compares the multiplication factors ($k_{eff}$) between ENDF/B-VIII.0 with DBRC and ENDF/B-VII.1 without DBRC. As might be expected, considering epithermal upscattering resulted in an additional negative reactivity bias of 150–200 pcm for the PWR hot full-power calculations. However, reactivities with epithermal upscattering were increased because of depletion history which is to follow up individual isotopes through isotopic depletion calculation with decays and neutron–nuclide reactions. Therefore, once the ENDF/B library is improved to ensure better reactivity at the beginning of burnup, low reactivity at high burnups may not be an issue.

The most influential nuclides in terms of impact on the reactivity difference between ENDF/B-VII.1 and VIII.0 were identified by Kim and Wieselquist [11]. Figure 2 shows the $k_{eff}$ differences as a function of burnup caused by the ENDF/B-VII.1 cross sections of all nuclides, $^{235}$U, $^{238}$U, $^{239}$Pu and $^{16}$O. There were error cancelations at zero burnup between $^{238}$U (+230 pcm), $^{235}$U (−210 pcm) and $^{16}$O (−120 pcm). As burnup increased and the $^{239}$Pu influence became more negative and then more saturated, the $^{238}$U influence decreased and became more negative above 40 MWD/kgU. The $^{235}$U influence decreased and became saturated at −100 pcm. The total $k_{eff}$ difference at 60 MWD/kgU was about 400 pcm, which was 650 pcm in $\Delta\rho$. More detailed reaction rate analysis was performed to demonstrate energy and reaction dependent reaction rate differences to impact the overall reactivities. This analysis is discussed in Section 4.

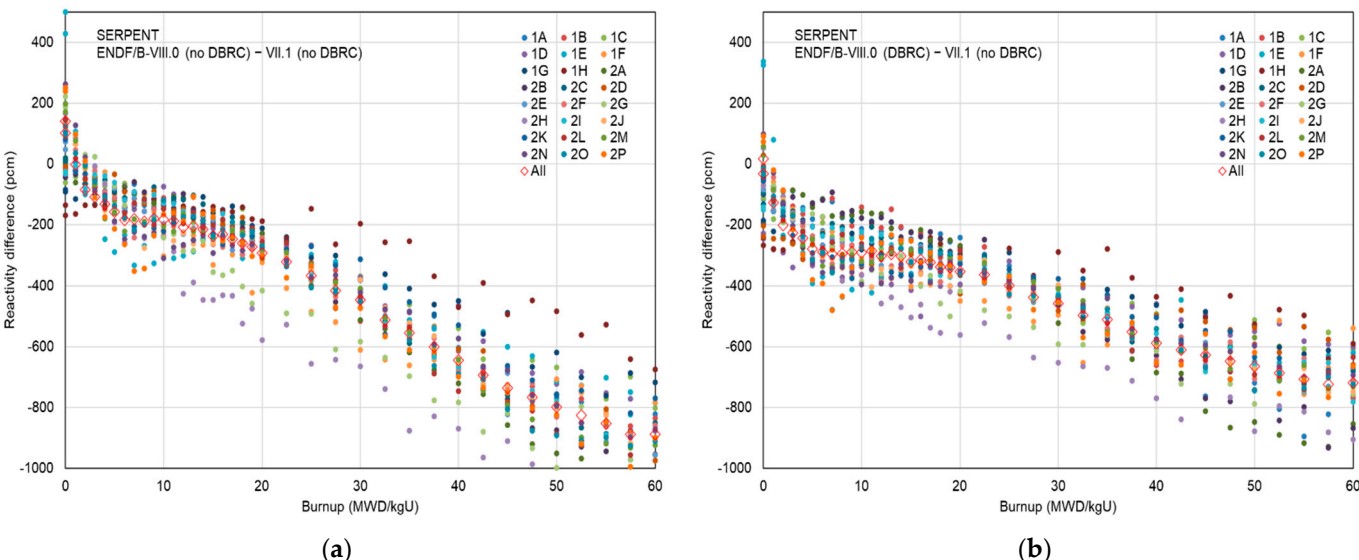

**Figure 1.** Comparison of the multiplication factors as a function of burnup: (**a**) the reactivity difference between ENDF/B-VIII.0 without DBRC and ENDF/B-VII.1 without DBRC; (**b**) the reactivity difference between ENDF/B-VIII.0 with DBRC and ENDF/B-VII.1 without DBRC.

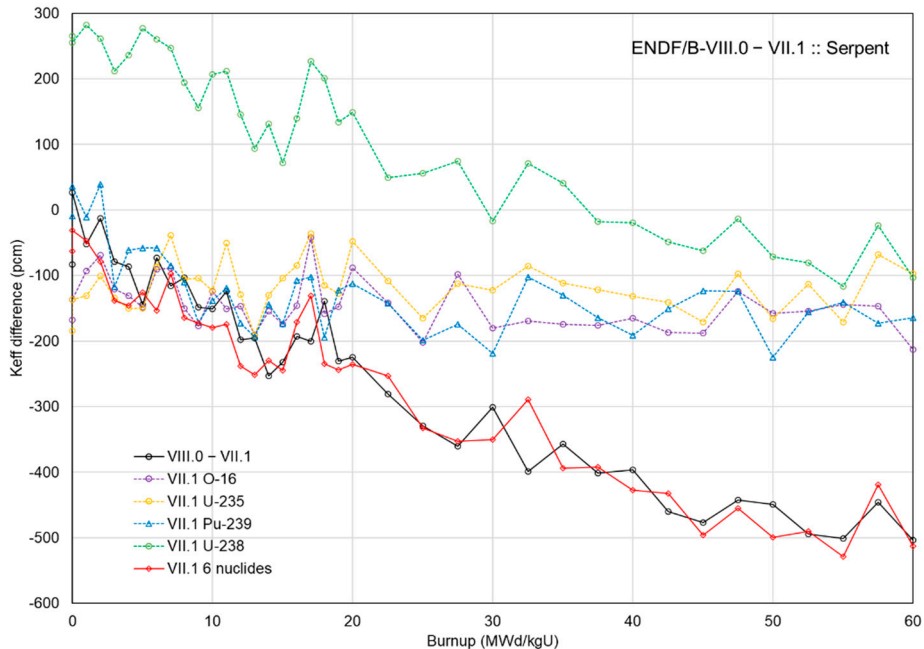

**Figure 2.** Most influential nuclides in ENDF/B-VIII.0.

### 3.1.4. Extensive PWR Pin and Assembly Benchmark Problems

The extensive PWR pin and assembly benchmark calculations were performed using SCALE/KENO with ENDF/B-VII.1 and -VIII.0, with and without considering epithermal upscattering. The histogram of the reactivity differences for the pin benchmark problems is shown in Figure 3a. ENDF/B-VIII.0 mostly underestimated reactivities, except for the $^{235}$U 2.1 wt % cases. The same benchmark calculations used for the extensive PWR pin benchmark problems were performed using MCNP. Although MCNP slightly overestimated reactivity, compared to KENO, MCNP's overall trend was very similar to KENO's. Tables 7 and 8 provide the benchmark results categorized for $^{235}$U enrichment, soluble boron concentration, moderator density and fuel temperature for fuel pins and assemblies, respectively. A trend of reactivity bias can be clearly observed in $^{235}$U enrichment and

soluble boron concentration. Under hot conditions, the boron worth of ENDF/B-VII.1 was larger than that of ENDF/B-VIII.0 by 0.093 pcm/ppm. There was no trend for moderator density and fuel temperature that would provide very consistent moderator and fuel temperature reactivity coefficients.

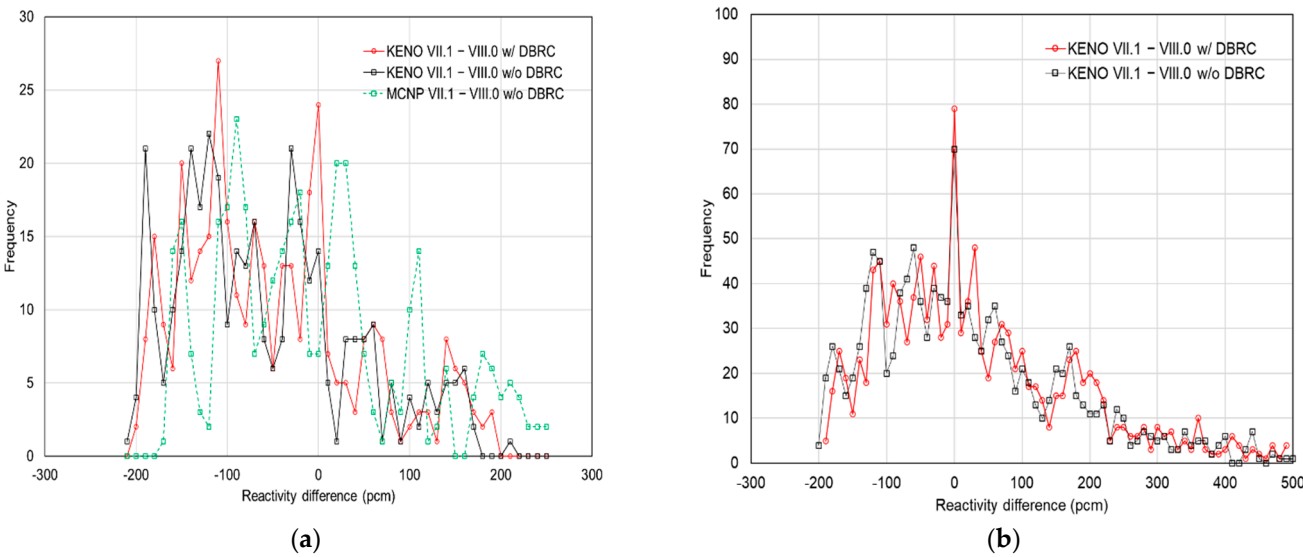

**Figure 3.** Histogram of the extensive PWR (**a**) pin and (**b**) assembly benchmark results.

**Table 7.** Results of the extensive PWR pin benchmark problems.

| | | Reactivity Differences between ENDF/B-VIII.0 and VII.1 (pcm) | | | | | | | | Counts |
|---|---|---|---|---|---|---|---|---|---|---|
| | | **KENO with DBRC** | | | | **KENO without DBRC** | | | | |
| | | **Average** | **S. Dev.** | **Min.** | **Max.** | **Average** | **S. Dev.** | **Min.** | **Max.** | |
| Total | All | −56 | 114 | −206 | 194 | −66 | 119 | −215 | 206 | 360 |
| | Hot | −51 | 113 | −206 | 194 | −62 | 118 | −215 | 206 | 324 |
| | Cold | −105 | 122 | −199 | 35 | −108 | 126 | −209 | 50 | 36 |
| **Hot** | | | | | | | | | | |
| $^{235}$U (wt %) | 2.1 | 61 | 94 | −47 | 194 | 50 | 86 | −60 | 206 | 108 |
| | 3.1 | −69 | 85 | −149 | 38 | −81 | 95 | −157 | 13 | 108 |
| | 4.1 | −145 | 149 | −206 | 0 | −154 | 158 | −215 | 0 | 108 |
| Boron PPM | 0 | −111 | 131 | −206 | 7 | −122 | 140 | −215 | 0 | 108 |
| | 600 | −53 | 100 | −165 | 101 | −63 | 106 | −179 | 80 | 108 |
| | 1300 | 11 | 105 | −132 | 194 | 1 | 103 | −145 | 206 | 108 |
| Density (g/cm$^3$) | 0.7408 | −46 | 113 | −205 | 194 | −57 | 118 | −212 | 172 | 108 |
| | 0.7032 | −50 | 111 | −198 | 177 | −62 | 117 | −203 | 206 | 108 |
| | 0.6560 | −56 | 114 | −206 | 188 | −66 | 119 | −215 | 164 | 108 |
| Fuel temp. (K) | 600 | −60 | 115 | −206 | 162 | −69 | 121 | −212 | 172 | 108 |
| | 900 | −52 | 113 | −194 | 185 | −60 | 115 | −205 | 164 | 108 |
| | 1200 | −41 | 110 | −200 | 194 | −55 | 117 | −215 | 206 | 108 |
| **Cold** | | | | | | | | | | |
| $^{235}$U (wt %) | 2.1 | −44 | 65 | −102 | 35 | −45 | 70 | −115 | 50 | 12 |
| | 3.1 | −112 | 118 | −160 | 0 | −116 | 121 | −173 | 0 | 12 |
| | 4.1 | −160 | 164 | −199 | 0 | −164 | 167 | −209 | 0 | 12 |
| Boron PPM | 0 | −150 | 155 | −199 | 0 | −156 | 161 | −209 | 0 | 12 |
| | 600 | −108 | 119 | −171 | 0 | −108 | 118 | −171 | 0 | 12 |
| | 1300 | −57 | 82 | −150 | 35 | −60 | 87 | −136 | 50 | 12 |

**Table 8.** Results of the extensive PWR assembly benchmark problems.

| | | Reactivity Differences between ENDF/B-VIII.0 and VII.1 (pcm) | | | | | | | | Counts |
|---|---|---|---|---|---|---|---|---|---|---|
| | | KENO with DBRC | | | | KENO without DBRC | | | | |
| | | Average | S. Dev. | Min. | Max. | Average | S. Dev. | Min. | Max. | |
| Total | All | 32 | 154 | −198 | 554 | 18 | 150 | −214 | 545 | 1260 |
| | Hot | 43 | 158 | −198 | 554 | 28 | 154 | −214 | 545 | 1134 |
| | Cold | −65 | 104 | −196 | 183 | −72 | 107 | −201 | 239 | 126 |
| **Hot** | | | | | | | | | | |
| $^{235}$U (wt %) | 2.1 | 178 | 227 | −39 | 554 | 160 | 213 | −52 | 545 | 378 |
| | 3.1 | 21 | 108 | −135 | 301 | 7 | 106 | −161 | 297 | 378 |
| | 4.1 | −71 | 109 | −198 | 190 | −84 | 119 | −214 | 132 | 378 |
| Boron PPM | 0 | −24 | 134 | −198 | 407 | −38 | 137 | −214 | 372 | 378 |
| | 600 | 40 | 148 | −154 | 488 | 26 | 142 | −181 | 469 | 378 |
| | 1300 | 111 | 188 | −125 | 554 | 96 | 179 | −134 | 545 | 378 |
| Density (g/cm$^3$) | 0.7408 | 46 | 159 | −197 | 536 | 32 | 154 | −209 | 545 | 378 |
| | 0.7032 | 43 | 158 | −198 | 554 | 27 | 153 | −213 | 519 | 378 |
| | 0.6560 | 39 | 158 | −198 | 524 | 25 | 154 | −214 | 508 | 378 |
| Fuel temp. (K) | 600 | 30 | 149 | −198 | 507 | 19 | 148 | −209 | 520 | 378 |
| | 900 | 44 | 160 | −190 | 554 | 28 | 153 | −214 | 545 | 378 |
| | 1200 | 54 | 165 | −198 | 550 | 36 | 159 | −209 | 510 | 378 |
| **Cold** | | | | | | | | | | |
| $^{235}$U (wt %) | 2.1 | −1 | 75 | −109 | 183 | −10 | 78 | −116 | 239 | 42 |
| | 3.1 | −71 | 95 | −161 | 154 | −78 | 97 | −173 | 59 | 42 |
| | 4.1 | −124 | 134 | −196 | 38 | −128 | 137 | −201 | 16 | 42 |
| Boron PPM | 0 | −118 | 132 | −196 | 59 | −126 | 138 | −201 | 73 | 42 |
| | 600 | −67 | 94 | −160 | 117 | −74 | 95 | −167 | 102 | 42 |
| | 1300 | −11 | 80 | −111 | 183 | −16 | 79 | −124 | 239 | 42 |

Figure 3b provides the distribution of the reactivity differences between ENDF/B-VIII.0 and -VII.1 for the extensive PWR assembly benchmark problems. Although ENDF/B-VIII.0 underestimated reactivity for the assemblies without any absorbers, it mostly overestimated reactivity for the assemblies with various burnable absorbers, such as IFBA, Pyrex and Gadolinia. $\Delta\rho$ differences of more than 500 pcm were observed in the WB2M fuel assemblies with 2.1 wt % $^{235}$U enrichment, including 128 IFBAs. Similar reactivity bias trends were observed for $^{235}$U enrichment and soluble boron concentration. The effect of $^{10}$B and Gd isotopes in increasing reactivity is discussed in Section 4. Because those nuclides would burn very quickly as a burnable absorber to control reactivity, they would not impact cycle length.

3.1.5. Seoul National University PWR Nonuniform Fuel Temperature Problems

Figures 4 and 5 compare temperature-dependent reactivities between ENDF/B-VII.1 and-VIII.0 for the Seoul National University benchmark problems with uniform and nonuniform temperature distributions, respectively, in fuel pellets with and without considering epithermal upscattering. The benchmark calculations were performed using KENO with the on-the-fly temperature interpolation capability. ENDF/B-VIII.0 always underestimated reactivities by about 150 pcm above hot full power conditions. However, because the slopes of ENDF/B-VII.1 and VIII.0 seem to be similar, fuel temperature reactivity coefficients would be expected to be similar with each other.

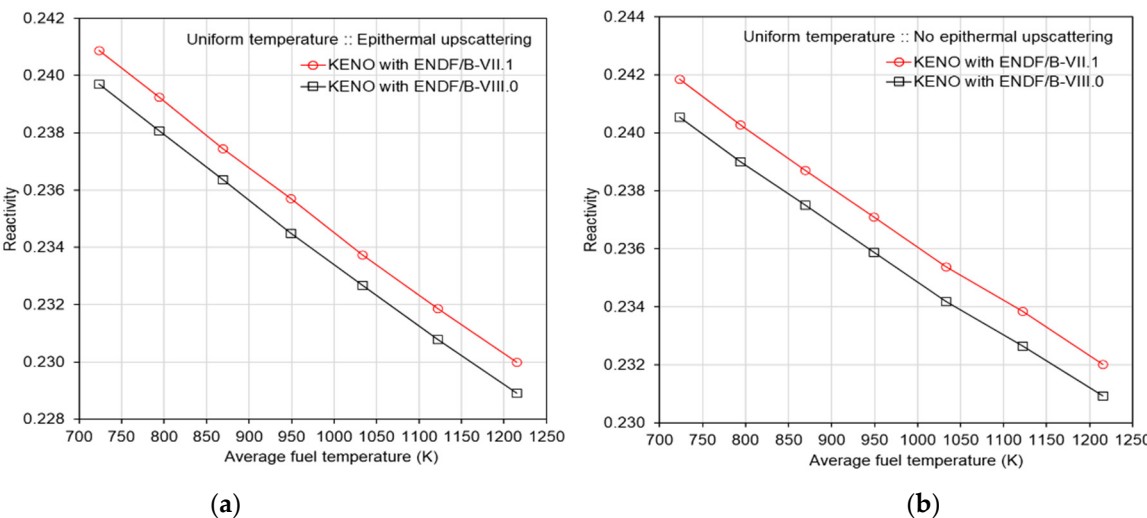

**Figure 4.** Comparison of reactivities with uniform fuel temperature distribution: (**a**) epithermal upscattering; (**b**) no epithermal upscattering.

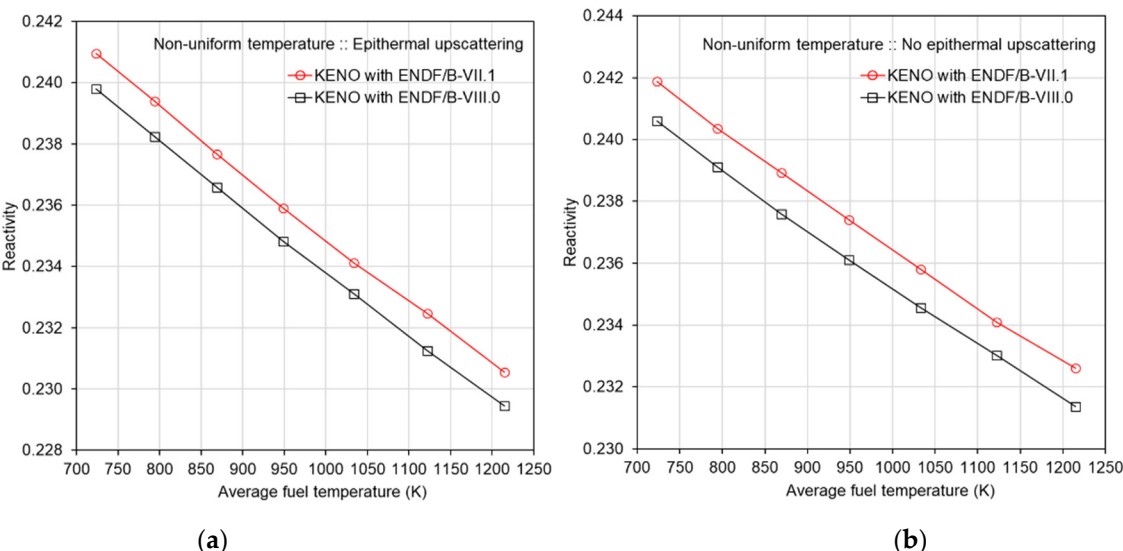

**Figure 5.** Comparison of reactivities with nonuniform fuel temperature distribution: (**a**) epithermal upscattering; (**b**) no epithermal upscattering.

### 3.1.6. PWR Mosteller Benchmark Problems

Figure 6 provides the results for the Mosteller benchmark problems. These results compare the Doppler temperature coefficients between ENDF/B-VII.1 and -VIII.0 for various $^{235}$U enrichments with and without considering epithermal upscattering. The benchmark calculations were performed using SCALE/KENO and MCNP. Good consistency was shown in the Doppler temperature coefficients between ENDF/B-VII.1 and -VIII.0, regardless of $^{235}$U enrichment. It has been shown that Doppler temperature coefficients considering epithermal upscattering are lower than those that do not consider epithermal upscattering by about 10–15%, which would ensure much better agreement with the measured plant data [9].

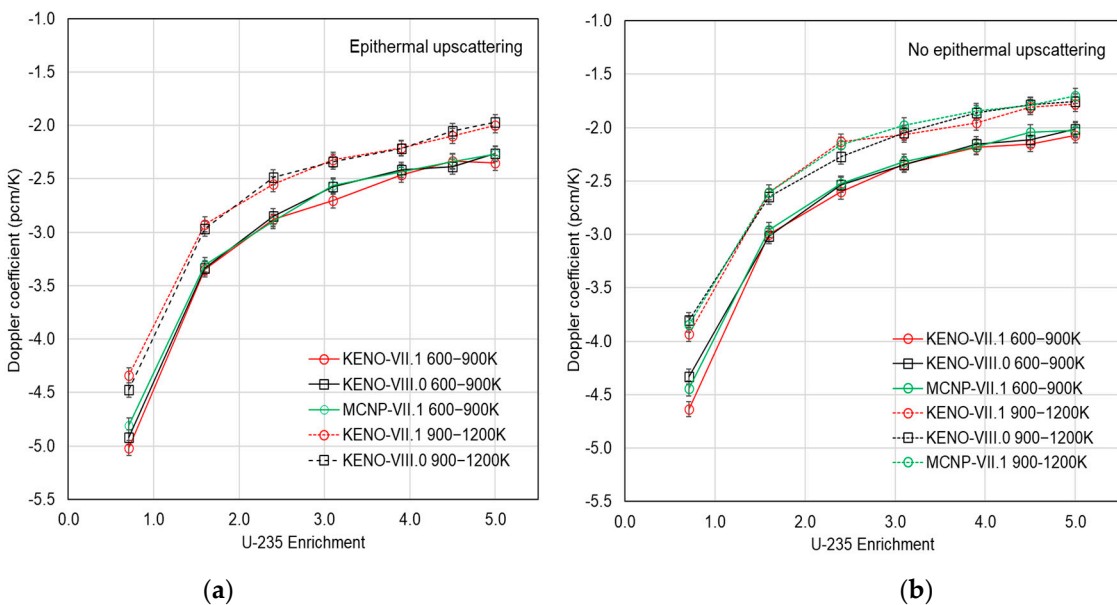

**Figure 6.** Comparison of Doppler temperature coefficients with various $^{235}$U enrichments: (**a**) epithermal upscattering; (**b**) no epithermal upscattering.

### 3.2. BWR Benchmark Results

### 3.2.1. BWR Depletion Benchmark Problems

The depletion trend of BWR fuels was expected to be similar to that of PWR fuels because their compositions and operation conditions are very similar, except for soluble boron in PWRs and void in BWRs. Therefore, single-pin investigation alone with various void fractions may be sufficient to identify the depletion characteristics for BWR fuel. Figure 7 provides the reactivity differences between ENDF/B-VIII.0 and -VII.1 for the BWR depletion benchmark problems. The selected BWR fuel pins included a much larger fuel pellet radius, which was 0.60579 cm; the PWR pellet radius was 0.4025 cm. The overall depletion trend for the BWR pins, 1A–1F, without void was very similar to the PWR depletion trend. The 1G and 1H cases are 50% and 70% voids, respectively; the reactivity differences at high burnups were significantly less than the reactivity differences with zero voids.

### 3.2.2. Extensive BWR Pin and Assembly Benchmark Problems

Extensive VERA BWR pin and assembly benchmark calculations were performed using SCALE/KENO and MCNP with ENDF/B-VII.1 and -VIII.0 only, without considering epithermal upscattering. The reactivity differences for the pin and assembly benchmark problems are shown in Figure 8. ENDF/B-VIII.0 mostly underestimated reactivities for the pin benchmark problems, but some reactivity overestimations were observed in the selected cases. Many more cases with reactivity overestimations were observed for the assembly benchmark problems, which were likely caused by the gadolinia burnable poisons.

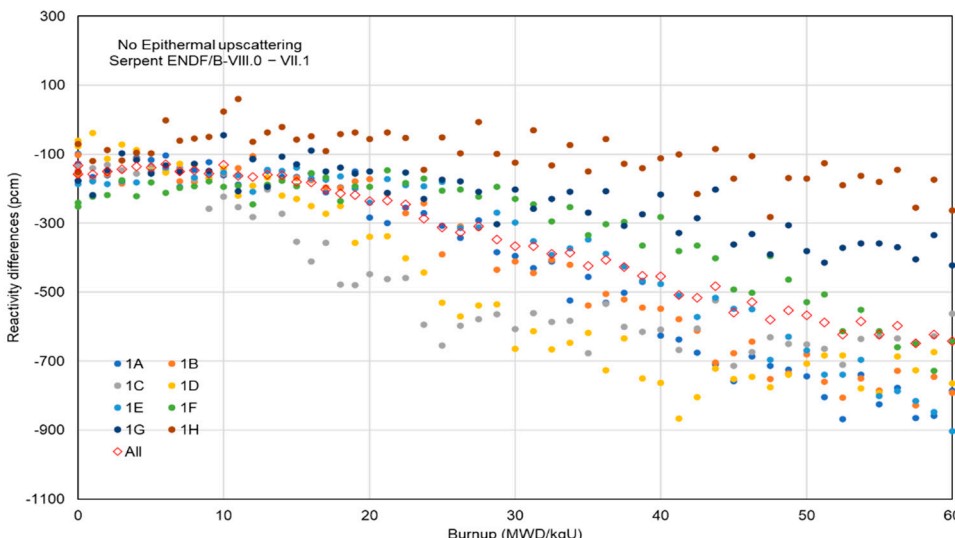

**Figure 7.** Comparison of the multiplication factors as a function of burnup for the BWR depletion benchmark problems without considering epithermal upscattering.

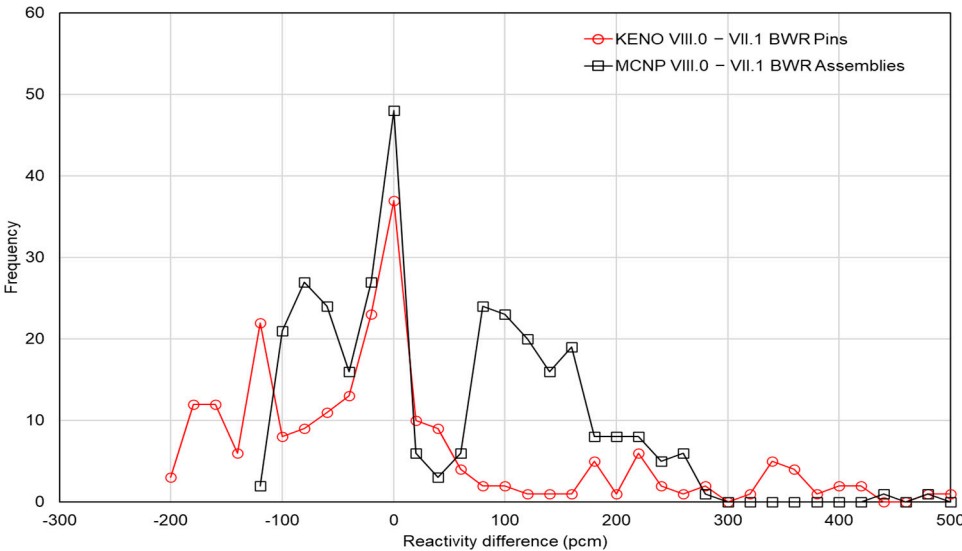

**Figure 8.** BWR extensive pin and assembly benchmark results.

Table 9 provides the benchmark results categorized for $^{235}$U enrichment, void fraction, burnup, control rod and fuel temperature. A trend of reactivity bias can be seen in the $^{235}$U enrichment and void fraction. $\Delta\rho$ differences of more than 400 pcm were observed in the 2.1 wt % $^{235}$U enriched pin with 90% void, a fuel temperature of 1200 K and zero burnup, as well as the mixed enriched BWR assemblies with 80% void and control rods inserted at 600 and 900 K fuel temperatures.

**Table 9.** Results of the BWR extensive pin and assembly problems.

| Reactivity Differences between ENDF/B-VIII.0 and VII.1 (pcm) | | | | | | | | | | | | | |
|---|---|---|---|---|---|---|---|---|---|---|---|---|---|
| Single pins | | | | | | | | | Assemblies | | | | |
| | | Average | S. Dev. | Min. | Max. | Counts | | | Average | S. Dev. | Min. | Max. | Counts |
| Total | All | −10 | 157 | −215 | 509 | 220 | Total | All | 35 | 119 | −132 | 474 | 320 |
| $^{235}$U (wt %) | 2.1 | 79 | 188 | −113 | 509 | 52 | $^{235}$U (wt %) | Mixed | 166 | 187 | −58 | 474 | 80 |
| | 3.1 | −20 | 119 | −174 | 411 | 116 | | 2.1 | 94 | 103 | −24 | 187 | 80 |
| | 4.1 | −78 | 192 | −215 | 363 | 52 | | 3.1 | −23 | 33 | −71 | 78 | 80 |
| | | | | | | | | 4.1 | −96 | 98 | −132 | −15 | 80 |
| Void (%) | 0 | −65 | 100 | −211 | 74 | 112 | Void (%) | 0 | 0 | 0 | 0 | 0 | 0 |
| | 50 | −106 | 130 | −215 | 34 | 36 | | 40 | 40 | 40 | 40 | 40 | 40 |
| | 70 | −44 | 91 | −160 | 118 | 36 | | 80 | 80 | 80 | 80 | 80 | 80 |
| | 90 | 291 | 306 | 134 | 509 | 36 | | - | - | - | - | - | - |
| Burnup (MWD/kgU) | 0 | −8 | 184 | −215 | 509 | 156 | Control rod | Out | 23 | 105 | −132 | 239 | 160 |
| | 10 | −59 | 63 | −102 | 0 | 16 | | In | 48 | 131 | −127 | 474 | 160 |
| | 20 | −9 | 24 | −52 | 31 | 16 | | - | - | - | - | - | - |
| | 40 | 12 | 38 | −43 | 74 | 16 | | - | - | - | - | - | - |
| | 60 | −8 | 39 | −70 | 58 | 16 | | - | - | - | - | - | - |
| Fuel temp. (K) | 293 | −92 | 112 | −204 | 35 | 28 | Fuel temp. (K) | 293 | −27 | 56 | −112 | 122 | 32 |
| | 600 | −6 | 155 | −211 | 422 | 64 | | 600 | 36 | 123 | −130 | 474 | 96 |
| | 900 | 3 | 165 | −215 | 479 | 64 | | 900 | 44 | 125 | −132 | 439 | 96 |
| | 1200 | 9 | 166 | −203 | 509 | 64 | | 1200 | 47 | 123 | −129 | 276 | 96 |

## 4. Discussion

Extensive benchmark calculations were performed using various CE MC codes, such as SCALE/KENO, Serpent and MCNP, with the ENDF/B-VII.1 and VIII.0 libraries to identify the neutronic characteristics of ENDF/B-VIII.0 compared to those of ENDF/B-VII.1 for LWR analysis. The underlined neutronic characteristics of ENDF/B-VIII.0 are summarized below.

**$^{235}$U enrichment reactivity bias**. Figure 3 shows that the $^{235}$U and $^{16}$O reactivity differences at zero burnup were −180 and −170 pcm, respectively, but the $^{238}$U reactivity difference was +260 pcm. Because the $^{10}$B reactivity difference was positive, effective reactivity differences were canceled out between them, so no significant excess reactivity differences were observed. Figure 9a provides the energy- and reaction-dependent reactivity differences for $^{16}$O, $^{235}$U and $^{238}$U, which were obtained by converting the reaction rate differences into the reactivity differences with the same neutron spectra. Because the excess reactivity of $^{235}$U absorption and production was negative, the difference of absorption was larger than that of production. Therefore, as $^{235}$U enrichment increased, the excess reactivity differences of both $^{235}$U and $^{238}$U was more negative, resulting in reactivity underestimation in ENDF/B-VIII.0.

**Positive reactivity of Gd isotopes**. Figure 10a compares the flux-averaged cross sections of $^{155}$Gd and $^{157}$Gd in a single gadolinia rod between ENDF/B-VII.1 and -VIII.0 in the 51-group structure. Even though there was no change in the pointwise cross sections of $^{155}$Gd and $^{157}$Gd, there were some differences in the flux-averaged MG cross sections due to different neutron spectra. The effective ENDF/B-VIII.0 MG cross sections at very thermal energy were smaller than those of ENDF/B-VII.1, which would result in positive reactivity, as observed in the PWR progression cases 2O and 2P.

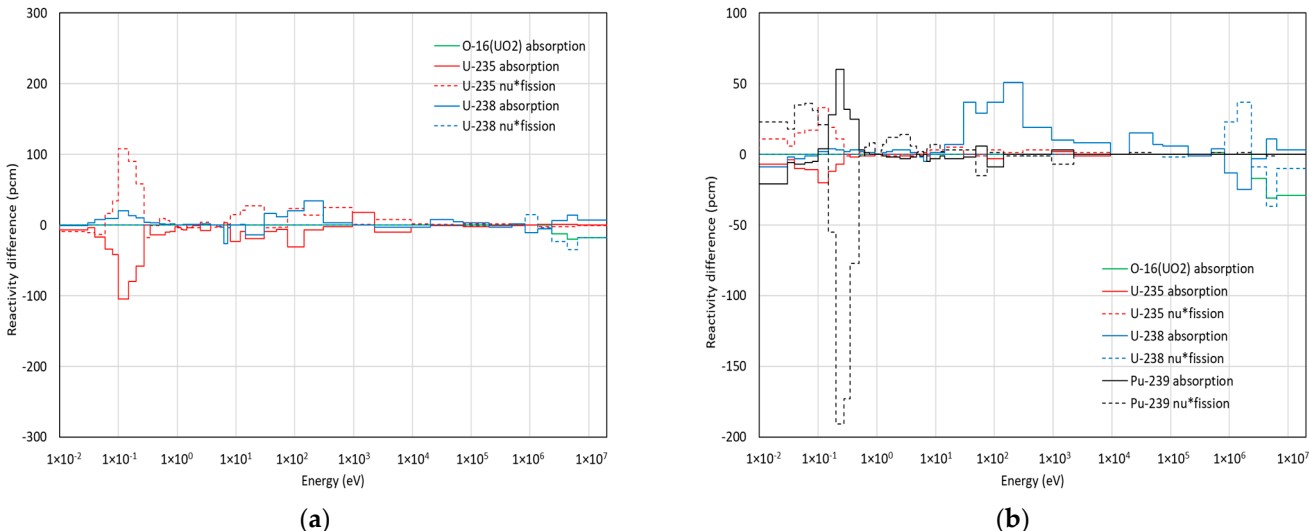

**Figure 9.** Reaction-dependent groupwise reactivity differences caused by the reaction rate differences for the typical PWR fuel pin: (**a**) zero burnup; (**b**) 60 MWD/kgU burnup.

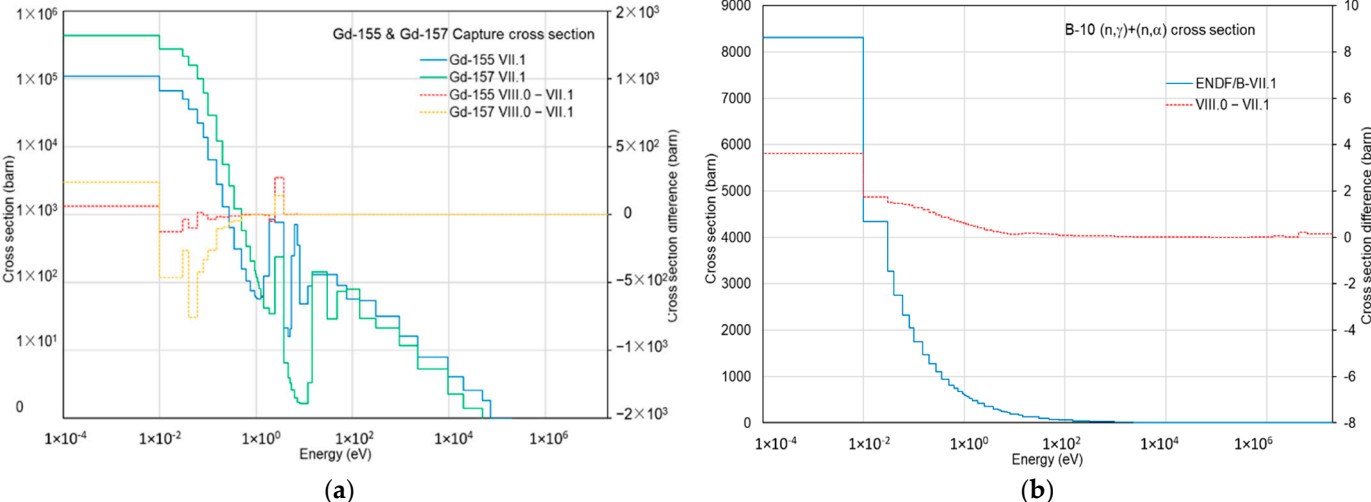

**Figure 10.** Comparison of the flux-weighted capture cross sections between ENDF/B-VII.1 and -VIII.0 in the 51-group structure: (**a**) $^{155}$Gd and $^{157}$Gd; (**b**) $^{10}$B.

**Positive reactivity of $^{10}$B**. Figure 10b compares the $^{10}$B absorption cross sections, including $(n, \gamma)$ and $(n, \alpha)$ reactions, between ENDF/B-VII.1 and -VIII.0 in the 51-group structure. The cross sections were flux-weighted using typical PWR pointwise spectra. The $^{10}$B capture cross sections of ENDF/B-VIII.0 were almost identical to those of ENDF/B-VII.1 at all energy ranges, which may not impact eigenvalues. However, the ENDF/B-VIII.0 $^{10}$B would result in less reactivity worth, which is common for soluble boron, IFBA, Pyrex and B$_4$C control rods—quite different from the positive cross-section difference. Figure 11a compares the ENDF/B-VII.1 and VIII.0 neutron spectra for a single pin with IFBA. ENDF/B-VIII.0 underestimated thermal and very fast neutron fluxes and it overestimated epithermal flux. The difference would result in significant reactivity increases for $^{235}$U, $^{238}$U and $^{10}$B, as shown in Figure 11b.

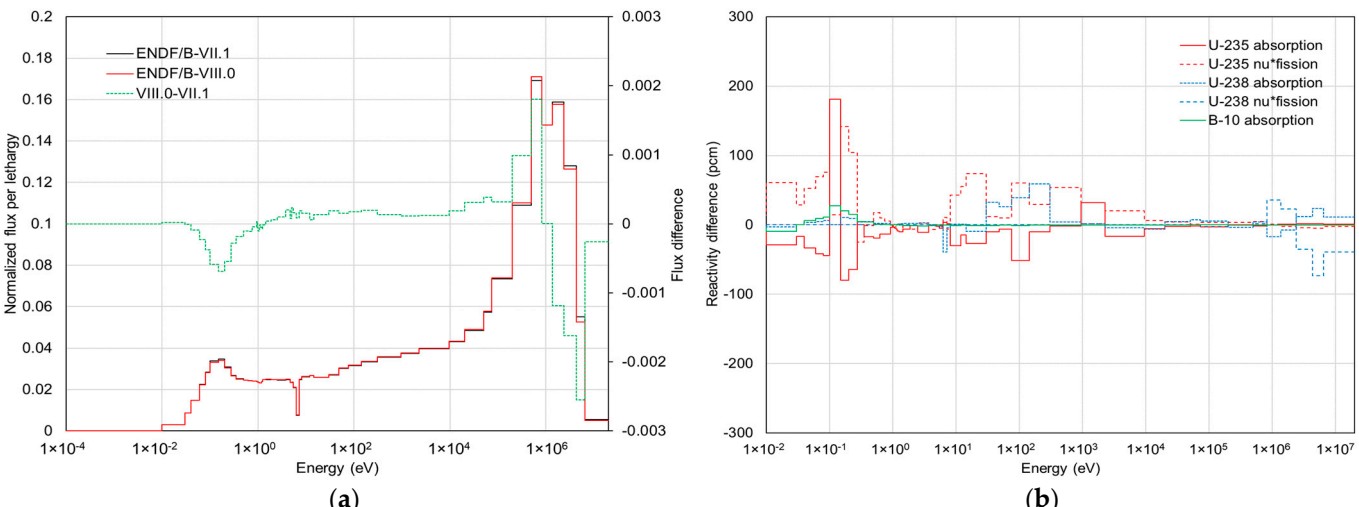

**Figure 11.** Comparison of the flux and reactivity for IFBA between ENDF/B-VII.1 and -VIII.0 in the 51-group structure: (**a**) neutron spectra; (**b**) reactivity differences.

**Depletion history effect** [11]. This effect is perhaps the most challenging issue in ENDF/B-VIII.0. When performing static and snapshot burnup calculations, no significant reactivity differences were observed. However, when considering depletion history, there was a significant reactivity underestimation in ENDF/B-VIII.0 that may limit its use in LWR analysis. The most influential nuclides included $^{239}$Pu, $^{235}$U, $^{16}$O and $^{238}$U.

**Positive reactivity of voids**. There was a relatively large trend for the BWR single pins with various voids. However, there was almost no trend for the BWR fuel assemblies, which may have been caused by error cancelation. Even though there was a large positive reactivity at 90% void, it may not impact BWR analysis.

**Epithermal upscattering**. Theoretically, epithermal upscattering is much more physical than no epithermal upscattering and it would introduce better agreement with the measured Doppler temperature coefficient. However, epithermal upscattering has not been considered properly in LWR analysis because of concerns about its reactivity underestimation. The ENDF/B-VII.1 PWR hot zero power results without considering epithermal upscattering were very consistent with the measured plant data. Considering epithermal upscattering may cause some reactivity differences in the measured-to-calculated comparison. However, the investigation in this study indicates that epithermal upscattering may not cause any issues in fuel cycle length, as shown in Figure 1b.

**Moderator and fuel temperature coefficients**. Because no sensitivity to the moderator density as provided in Tables 7 and 8 was observed, there was no impact on moderator temperature coefficients in ENDF/B-VIII.0. As shown in Figures 5 and 6, there was also no sensitivity to fuel temperature.

**Fission product nuclides**. ENDF/B-VIII.0 cross-section differences in fission product yield nuclides had almost no impact on reactivity.

**Decay and fission product yield data** [11]. The ENDF/B-VIII.0 decay and fission product yield data would not impact the multiplication factors. It is noted that the fission product yield data were adopted from ENDF/B-VII.1, which came from England and Ryder's seminal work from around 1990 with some modest updates for Plutonium fast neutron fission product yield data for ENDF/B-VII.1 [10].

**Thermal scattering data for $^1$H in H$_2$O**. Figure 12 compares neutron spectra and group-wise reactivities for a typical PWR fuel pin between the ENDF/B-VIII.0 and VII.1 $^1$H S($\alpha,\beta$) thermal scattering data in which the ENDF/B-VIII.0 data were used for other nuclides. Eigenvalue was underestimated with the ENDF/B-VII.1 $^1$H S($\alpha,\beta$) data by about 60 pcm. There was some change in thermal neutron spectrum in fuel as shown in Figure 12a which would result in reactivity differences in the $^{235}$U absorption and fission reactions.

Figure 12b shows error cancellations between the absorption and fission reactions, but there was overall positive reactivity with the ENDF/B-VIII.0 $^{1}$H S($\alpha,\beta$) thermal scattering data.

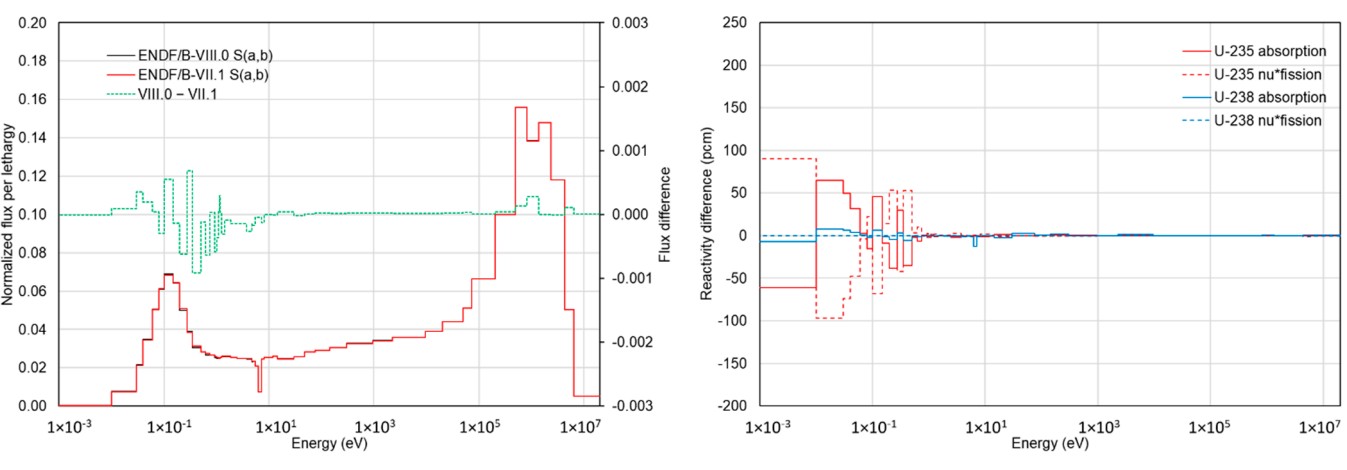

**Figure 12.** Comparison of the flux and reactivity between the ENDF/B-VIII.0 and VII.1 $^{1}$H S (**a**,**b**) thermal scattering data in the 51-group structure: (**a**) neutron spectra; (**b**) reactivity differences.

## 5. Conclusions

Neutronic characteristics of ENDF/B-VIII.0 neutron cross-section data were investigated through a comparison with ENDF/B-VII.1. This type of investigation may need to be completed during evaluation of new ENDF/B data and there should be a collaborative iteration between the nuclear data evaluators and reactor physicists. The authors propose that the most important issues to resolve for ENDF/B-VIII.1 are as follows:

- The depletion history effect should be eliminated, where ENDF/B-VIII.0 data have shown a significant increase in burnup-dependent reactivity bias compared to ENDF/B-VII.1. Six nuclides were responsible for a bias that essentially starts at 0 pcm for fresh fuel (due to cancellation of errors among the six nuclides) and decreases linearly to -800 pcm $\Delta\rho$ at a burnup of 60 GWd/MTU. This underprediction of reactivity with ENDF/B-VIII.0 is noticeable in comparisons with power plant data and essentially prevents ENDF/B-VIII.0 from being endorsed for LWR simulations.

- The epithermal scattering issue should be revisited, which exists in both ENDF/B-VII.1 as well as VIII.0. A higher-fidelity physics model is not used in practice because it introduces additional bias when comparing with measured data. Although, due to compensating effects, such an occurrence is not a surprise, this particular effect should be understood better. It may be that a new high-precision measurement is required to prove unequivocally the value of the higher-fidelity epithermal scattering treatment, at which point it could be used as the default in simulations and we can turn our attention to reducing the compensating errors.

Even though new ENDF/B data are developed through validation with the experiment data, data availability is very limited, so nuclear data issues in reactor physics analysis may not be effectively addressed in finalizing new ENDF/B libraries.

Many measured data from power plants across various advanced reactor systems exist. However, those measured data cannot be effectively used in evaluating ENDF/B libraries. The authors suggest that extensive benchmark calculations be performed to evaluate expected behavior and impacts for new ENDF/B versions under development through comparisons between other versions of ENDF/B libraries. The authors also suggest developing a systematic strategy and procedure to ensure that power plant measured data are engaged in nuclear data evaluation. This would shorten the development period for new ENDF/B versions and minimize potential issues in new ENDF/B releases.

**Author Contributions:** Conceptualization, K.-S.K.; formal analysis, K.-S.K.; investigation, K.-S.K.; writing—original draft preparation, K.-S.K.; writing—review and editing, W.A.W.; supervision, W.A.W.; project administration, W.A.W.; funding acquisition, W.A.W. All authors have read and agreed to the published version of the manuscript.

**Funding:** This research was funded by the US Nuclear Regulatory Commission Office of Research with the agreement number 31310019N0008 for the SCALE code development, assessment and maintenance.

**Institutional Review Board Statement:** Not applicable.

**Informed Consent Statement:** Not applicable.

**Data Availability Statement:** The data presented in this study are available on request from the corresponding author.

**Acknowledgments:** This research was supported by the US Nuclear Regulatory Commission Office of Research and the US Department of Energy NEAMS project.

**Conflicts of Interest:** The authors declare no conflict of interest.

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
