# Peer review of "Neutronic Characteristics of ENDF/B-VIII.0 Compared to ENDF/B-VII.1 for Light-Water Reactor Analysis"

_jne, doi:10.3390/jne2040026_

Round 1

Reviewer 1 Report

I have no objections for the publication of this article.

Author Response

There was no request from reviewer.

Reviewer 2 Report

Enclosed some comments on your very good paper.

Sincerly

Author Response

Attached is authors' note to reviewer. 

Reviewer 3 Report

The paper documents a thorough examination of the differences between the ENDF/B-VII.1 and ENDF/B-VIII.0 nuclear data libraries for nuclear reactor simulations. It is well written and interesting to read. The members of the CSEWG community would be very interested in the results as they are in the midst of preparing ENDF/B-VIII.1 and would like to make appropriate corrections.

While the comparison between the two libraries is done well, the analysis only shows that the libraries are different. It would be nice to see how ENDF/B-VIII.0 performs as compared to the measured data that the authors repeatedly state are widely available. 

A few items that ought to be improved:

  • The reference to MCNP is quite old. Please use a reference for the version of MCNP that was used.
  • Section 3.1.3 states that ACE format libraries were used. Where were these obtained? Please provide appropriate citation(s).
  • The figures are quite difficult to read. There is a lot of information presented, but the figures are so small. Please make the figures bigger and use colors that are more contrasting so it is easier to see what is presented.

Author Response

Attached is authors' note for reviewer.

Reviewer 4 Report

The paper presents a very comprehensive analysis of the performance of the ENDF/B-VIII.0 library compared to its predecessor ENDF/B-VII.1. Not only do the authors compare the results but also try to identify the cause of the degraded performance, which can be of great value to evaluators, who are working on further improvements to evaluated data libraries. However, there are noted some errors in the paper, which most likely do not affect significantly the overall conclusions but nevertheless shed some doubt on the analysis and should be clarified.

I strongly agree with the authors that performance of any new library should be checked on real operating reactors including the performance with depletion, but one of the problems is that such data are mostly proprietary, in addition to requiring rather complex modelling effort. Some work (and will) would be needed to compile and release suitable data in the open literature (or data base) for easy implementation by the evaluators, who might not be top experts in modelling elaborate reactor systems with burnup. However, this is outside the scope of the present paper.

Overall, the paper is very good and useful to the community, but the errors mentioned below should be corrected.

General comments:

The curves in Figure 10 for both, Gd and B-10 are wrong. Both isotopes of Gd are nearly 1/v absorbers at thermal energies and the capture cross sections are identical in both libraries. The plot and the conclusions about the impact of the apparent differences should be corrected. Also, it is not clear what is plotted for B-10. The main contribution to absorption in B-10 is the (n,alpha) reaction, which is five orders of magnitude bigger than the capture cross section (n,gamma). B-10 is a distinct 1/v absorber and the (n,alpha) cross section is a standard. The difference between ENDF/B-VIII.0 and ENDF/B-VII.1 is only due to the change of Standards and amounts to 0.04%, which is unlikely to affect the calculations to any significant amount. The plot and the comment regarding the difference should be corrected.

The statement (line 352) that "ENDF/B-VIII.0 decay and fission product yield data would not impact the multiplication factor" is true: one should note that the fission product yield England & Ryder evaluations were adopted unchanged for ENDF/B-VIII.0. Perhaps the authors could make a note of this.

The majority of benchmarks refer to hot-full-power conditions. The analysis focuses on the cross sections and I agree that U-235, U-238 O-16 and Pu-239 are probably the most important nuclides, but elevated temperatures shift the thermal spectrum to higher energies, where thermal scattering law (TSL) data could play a role. For example, a substitution analysis (swapping ENDF/B-VII.1 TSL into the calculation with ENDF/B-VIII.0) data for at least one case would be useful, to exclude the possibility of a root cause for discrepancies in the TSL data.

Specific comments:

The fonts in all Figures are too small, especially those where two plots are shown side by side. The font size should be increased to be comparable to the font size in the text.

Most of the terms used in the paper are well defined or described, but what are "snapshot burnup calculations" and "depletion history" calculations (line 329-330)?

Author Response

Attached is authors' note to reviewer.

Round 2

Reviewer 4 Report

The improvements to revision-2 of the paper by the authors is very much appreciated and contributes to the overall clarity and soundness of the paper.